# Nutritional Knowledge and Dietary Practice in Elite 24-Hour Ultramarathon Runners: A Brief Report

**DOI:** 10.3390/sports7020044

**Published:** 2019-02-16

**Authors:** Roberto Citarella, Leila Itani, Vito Intini, Gregorio Zucchinali, Stefano Scevaroli, Dima Kreidieh, Hana Tannir, Dana El Masri, Marwan El Ghoch

**Affiliations:** 1CTR Centre of Rehabilitation Therapy, Reggio Emilia, Via Fratelli Cervi, 59/e, 42124 Reggio Emilia, Italy; citarella@ctr-re.it; 2Italian Ultra marathon and Trail Association, Via Moscardo, 47, 37142 Verona, Italy; vito.intini@gmail.com (V.I.); presidenza@iutaitalia.it (G.Z.); segreteria@iutaitalia.it (S.S.); 3Department of Nutrition and Dietetics, Faculty of Health Sciences, Beirut Arab University, P.O. Box 11-5020 Riad El Solh, Beirut 11072809, Lebanon; l.itani@bau.edu.lb (L.I.); d.kraydeyeh@bau.edu.lb (D.K.); hana.tannir@bau.edu.lb (H.T.); dana.masri@bau.edu.lb (D.E.M.)

**Keywords:** nutritional knowledge, dietary adequacy, ultramarathon, sporting performance, sport injury

## Abstract

Several factors contribute to athletes’ sporting performance and diet is a key component. Higher levels of nutritional knowledge seem to correlate with a higher quality of diet, but this remains poorly explored and findings are still not conclusive. The aim of our study was to assess nutritional knowledge and dietary adequacy, detecting any potential association between these two factors in elite 24-hour ultramarathon runners, a sport which seems to have been increasing in popularity over the last decade. Nutritional knowledge and Mediterranean dietary adequacy scores were assessed by means of validated questionnaires given to 10 elite ultramarathon runners (six males and four females) from the Italian Ultramarathon and Trail Association (IUTA). The overall nutritional knowledge in the entire sample of athletes seemed to be good, especially in terms of “dietary recommendations” and “nutrient sources” knowledge. However, females had higher total nutritional knowledge scores when compared to males. Finally, linear regression analysis showed that greater nutritional knowledge was positively associated with an increase in Mediterranean dietary adequacy scores (*β* = 1.27; 95% CI = 0.039–2.494; *p* = 0.045) after adjusting for level of education. Our findings provide evidence that higher nutritional knowledge is associated with better dietary practice in elite 24-hour ultramarathon runners. Future studies are needed to assess the usefulness of educational programs as a strategy to improve the adequacy of dietary intake in this specific population.

## 1. Introduction

Nutrition is an overlooked factor in athletes, despite its importance during sporting performance (i.e., training and competition), post-exercise recovery and in the prevention of risk of injury [1,2]. Moreover, it has been shown that some athletes previously had inadequate diets in terms of energy intake and nutrient needs [3,4]. Therefore, the identification of strategies that can improve dietary intake in this under-represented population is particularly crucial [5]. 

In the general population, a positive association has been widely reported between nutritional knowledge and dietary intake, with the former seeming to be a pivotal factor influencing the latter [6]. In fact, higher levels of knowledge are correlated with better diet quality [7]. However, surprisingly, this remains relatively unexplored in athletes [8]. Factors associated with this important shortcoming were evidenced by Heaney and colleagues in a recent systematic review, where they claimed that many studies use poor-quality and unvalidated tools to assess nutritional knowledge in athletes [8]. 

The 24-hour ultramarathon is one of the most demanding competitive sports in terms of energy expenditure and nutritional requirements during the competition as well as in post-exercise recovery [9,10]. Despite this fact, no investigations have addressed nutritional knowledge and dietary adequacy in this sport, which is becoming more popular worldwide. 

These considerations prompted us to assess nutritional knowledge by means of a validated questionnaire and to evaluate the usual dietary intake during the year, rather than that during peak training or recovery from a major race, by means of a dietary adequacy index in elite 24-hour ultramarathon runners. In addition, to extend the scope of this study, a potential association between nutritional knowledge and dietary adequacy was explored to underline the usefulness of implementing an educational nutritional program as a strategy that may improve the adequacy of dietary intakes in this specific population. We hypothesize the existence of an association between nutritional knowledge and dietary practices in this specific population.

## 2. Experimental Section

A nutritional assessment protocol was designed in collaboration with the Italian Ultramarathon and Trail Association (IUTA) (Italy), the Center of Rehabilitation Therapy, Reggio Emilia (Italy) and Beirut Arab University (Lebanon). The protocol involved elite runners from the IUTA national team who were interested in participating in the study. No exclusion criteria were defined and participants were healthy with no contraindications for ultramarathon running. After being informed about the protocol procedures, 10 athletes (six males and four females) agreed to participate in the study during their off-season period. The study was designed, reviewed and approved by the Institutional Review Board of IUTA (no. IUTA-2018-1111) and all participants gave informed consent in writing.

Body weight was measured to the nearest 0.1 kg in the clinical sample, using a mechanical weighing scale (SECA 756-7021009, Germany), and height was measured to the nearest 0.5 cm using the incorporated stadiometer. The body mass index (BMI) of each participant was then determined according to the standard formula of body weight (kg) divided by height (m) squared.

General questionnaire: This was administered to participants in order to retrieve information regarding their medical history and demographic and social conditions such as age, occupation, education, family income and marital status. 

The questions regarding the medical history were as follows. (i) Do you suffer from hypertension, cardiovascular diseases, hypercholesterolaemia, diabetes, osteoporosis, articular diseases, depression, anxiety or other specific diseases? (ii) If yes, do you take any medication for any of these diseases? (iii) In general, how would you assess your health status? In cases of an affirmative answer to the first two questions or a “poor” self-reported health status, a visit by the IUTA medical staff was scheduled to obtain more detailed information. 

Nutritional knowledge questionnaire: The questionnaire developed by Moynihan and colleagues is considered a simple tool that is quick to administer and easy to use in clinical practice [11]. Since the original version of the questionnaire was developed for a British population rather than an Italian one, at least as far as eating habits are concerned, cultural adaptation of the Italian validated version was carried out [12]. The final questionnaire consisted of 15 items and had four main subscales: (i) dietary recommendations; (ii) nutrient sources; (iii) healthiest meal option; and (iv) associations between diet and disease [12]. The total nutritional knowledge score was converted to a percentage and an average ≥50% was assumed to be satisfactory [13].

Mediterranean dietary adequacy score (MDAS): This is a dietary index that includes eight components representing the major food groups specified by the Mediterranean diet and its dietary guidelines [14]. The components and scoring procedures of the MDAS were determined a priori, with each component being scored from 0 (lowest adequacy) to 10 (highest adequacy). Because the food frequency questionnaire used in this study did not capture the portion size and number of portions, each eating occasion was assumed to represent the consumption of one portion [15]. The MDAS scoring system is reported in Table 1.

Descriptive statistics were calculated, i.e., means, standard deviations, frequencies and proportions where appropriate, to describe sociodemographic characteristics, nutritional knowledge, frequency of food intake and MDAS. A two-tailed independent sample *t*-test was conducted to compare mean age, BMI and intake for each of the MDAS components, with gender as an independent variable. A chi-square test was used to show that the frequencies across level of education, family income, occupation and self-rated health were independent of gender. Pearson’s correlation coefficient and simple and multiple linear regression analyses were used to study the association between MDAS and nutritional knowledge among the study participants. A scatter plot was used to illustrate the association. All analyses were done using SPSS version 25 (IBM Corp.; released 2017; IBM SPSS Statistics for Windows, version 25.0. Armonk, NY: IBM Corp). Statistical significance was considered at *p* < 0.05.

## 3. Results

Table 2 shows the age, BMI and sociodemographic characteristics of the study sample, with mean age = 41.0 ± 7.59 years and mean BMI = 21.66 ± 1.11 kg/m^2^. Most participants were employed, with a medium income and a good health status, with similar distributions for men and women across the categories of the dependent variables (Table 2). 

Females in the sample had higher Mediterranean dietary adequacy scores compared to their male counterparts (39.94 ± 6.33 vs. 57.50 ± 10.78, *p* < 0.05), with a higher score on the whole grains component (3.33 ± 5.16 vs. 10.00 ± 0.00, *p* < 0.05) and a lower score on the eggs component (2.71 ± 3.00 vs. 7.50 ± 0.00, *p* < 0.05) (Table 3). 

For nutritional knowledge, the score for the overall sample was good (77.50 ± 16.89%), with only one male participant scoring below 50%. Females scored significantly higher on the nutritional knowledge total score compared to males (92.36 ± 1.39% vs. 67.59 ± 14.77%), due to higher “diet–disease association” knowledge. They did not differ for other components of the nutritional knowledge score (Table 4). 

Correlation analysis revealed a significant association between the MDAS and total nutritional knowledge score (*r* = 0.675, *p* < 0.05). This association was confirmed by linear regression after adjusting for level of education (*β* = 1.27; 95% CI 0.039–2.494; *p* = 0.045), where every one-point increase in the nutritional knowledge score increased the MDAS by 1.27 points. Figure 1 illustrates the association between the two variables. 

## 4. Discussion

The present study aimed to provide benchmark data on nutritional knowledge and dietary intake adequacy in the Italian national 24-hour ultramarathon team and two major findings were revealed. 

Firstly, good overall “nutritional knowledge” was found in elite 24-hour ultramarathon runners, especially in terms of “dietary recommendations” and “nutrient sources” knowledge. Lower scores were reported on “healthiest meal option” and “diet–disease association” knowledge. The level of nutritional knowledge among athletes has been a question for researchers and previous findings are equivocal. On the other hand, this study found significantly greater nutritional knowledge in female ultramarathon runners than in males. This finding is in line with some previous studies [16,17] but contrasts with others, which found no difference in nutritional knowledge based on gender [18]. 

Secondly, this study found a strong positive association between nutritional knowledge and dietary adequacy in the sample. This finding has important implications, since better nutritional knowledge may have an impact on more adequate nutritional intake, known to be a key factor in sporting success. Several studies have assessed the association between athletes’ nutritional knowledge and their dietary intakes, and the findings of these studies vary [17,19,20]. Heaney and colleagues, in a systematic review, reported an accumulative, weak positive association between knowledge and dietary intake. However, they underlined some common flaws in these articles, including the use of inadequate or non-validated questionnaires [8]. In fact, the authors concluded in their systematic review that the nutritional knowledge of athletes and its impact on their dietary intake is equivocal [8].

This study has certain strengths. Most importantly, it is the first to assess “nutritional knowledge” and “Mediterranean dietary intake adequacy” among elite 24-hour ultramarathon runners. In fact, as far as we are aware, no study has been conducted on the issue in this specific population. Secondly, a validated Italian version of the questionnaire was used to assess nutritional knowledge, which eliminates any concerns about the questionnaire’s suitability for the scope. However, the study does have certain limitations. First and foremost, the sample size was too small, but this limitation is common in studies of elite athletes, as it is extremely difficult to recruit a large sample of elite 24-hour ultramarathon runners. Secondly, the cross-sectional design of this study should be considered another limitation, since the association found in this type of design cannot elicit information regarding any causality. Thirdly, there is no assessment of the daily energy, macronutrient and micronutrient intakes to assist in understanding what ultramarathon runners eat regularly. Finally, the researchers relied on self-reporting rather than objective measures of dietary intake assessment, as the questionnaire was not specifically designed for nutritional knowledge relating to sport. 

## 5. Conclusions

Our findings provide evidence that higher nutritional knowledge is associated with better dietary practice in elite 24-hour ultramarathon runners. Thus, education for the purpose of increasing knowledge may have beneficial outcomes for actual dietary practices. However, future studies are needed to assess the usefulness of educational programs as a strategy to improve the adequacy of dietary intake in this specific population.

## Figures and Tables

**Figure 1 sports-07-00044-f001:**
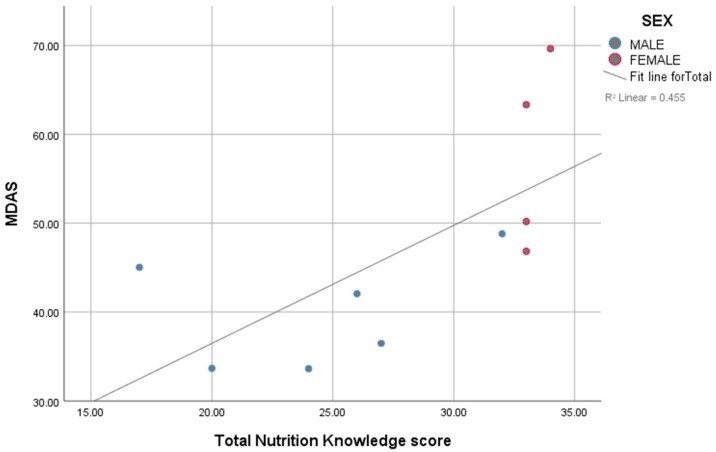
Simple scatter plot with fit line for Mediterranean dietary adequacy score against total nutrition knowledge score, by gender.

**Table 1 sports-07-00044-t001:** Calculation of the Mediterranean dietary adequacy score.

Components	Criteria for Minimum Score (0)	Criteria for Maximum Score (10)
Whole grains	No	Yes
Fruit and vegetables, portion per day	0	≥5
Legumes, portion per day	0	>2
Fish, portion per week	0	≥3
Red meat, portion per day	≥1.5	0
Eggs, per week	>4	0–4
Bakery, portion per day	>1	≤1
Alcohol, drink per day	-	-
Men	>2	0
Women	>1	0
Maximum score	0	80

**Table 2 sports-07-00044-t002:** Characteristics of the study sample (*n* = 10).

Sample Characteristics	Total (*n* = 10)	Males (*n* = 6)	Females (*n* = 4)	Significance	Statistic	Cohen’s *d*/Cramer’s *V*
Age (years)	41.0 ± 7.59	42.17 ± 5.88	39.50 ± 10.47	*p* = 0.666	*t* = 2.67	1.74
Body mass index (BMI) (kg/m^2^)	21.66 ± 1.11	21.79 ± 1.06	21.45 ± 1.31	*p* = 0.679	*t* = 34.0	0.22
Level of education	-	-	-	*p* > 0.05	X^2^ = 1.67	0.41
Intermediate	0 (0%)	0 (0%)	0 (0%)	-	-	-
High school	5 (50.0%)	2 (33.3%)	3 (75.0%)	-	-	-
University or higher	5 (50.0%)	4 (66.7%)	1 (25.0%)	-	-	-
Occupation	-	-	-	*p* > 0.05	X^2^ = 0.77	0.28
Employed	7 (70.0%)	4 (66.7%)	3 (75.0%)	-	-	-
Professional	2 (20.0%)	1 (16.7%)	1 (25.0%)	-	-	-
Self-employed	1 (10.0%)	1 (16.7%)	0 (0%)	-	-	-
Family income	-	-	-	*p* > 0.05	X^2^ = 1.67	0.41
Low (0–36,152 €)	1 (10.0%)	0 (0%)	1 (25.5%)	-	-	-
Medium (36,152–70,000 €)	9 (90.0%)	6 (100.0%)	3 (75.0%)	-	-	-
High (70,000–100,000 €)	0 (0%)	0 (0%)	0 (0%)	-	-	-
Self-reported health	-	-	-	*p* > 0.05	X^2^ = 0.104	0.10
Good	8 (80.0%)	5 (83.3%)	3 (75.0%)	-	-	-
Satisfactory	2 (20.0%)	1 (16.7%)	1 (25.0%)	-	-	-
Poor	0 (0%)	0 (0%)	0 (0%)	-	-	-

**Table 3 sports-07-00044-t003:** Mediterranean dietary adequacy scores among study participants (*n* = 10).

Components	Mediterranean Dietary Adequacy Score (MDAS)					
	Males	Females	Total	*p*-value	*t*-statistic	Cohen’s *d*
Whole grains	3.33 ± 5.16	10 ± 0.00	6.00 ± 5.16	<0.05	−3.16	−2.05
No	-	-	-	-	-	-
Yes	-	-	-	-	-	-
Fruit and vegetables, portion/day	3.07 ± 2.69	4.57 ± 3.81	3.67 ± 3.07	>0.05	−0.68	−0.44
Legumes, portion/day	1.37 ± 0.73	4.55 ± 3.81	2.64 ± 2.80	>0.05	−1.65	−1.07
Fish, portion/week	6.94 ± 3.40	7.50 ± 3.19	7.16 ± 3.15	>0.05	−0.26	−0.17
Red meat, portion/day	7.23 ± 1.55	8.69 ± 0.71	7.81 ± 1.44	>0.05	−2.02	−1.31
Eggs/week	2.71 ± 3.00	7.50 ± 0.00	4.63 ± 3.34	<0.05	−3.91	−2.54
Bakery, portion/day	5.36 ± 4.23	6.25 ± 5.41	5.65 ± 4.33	>0.05	−0.25	−0.16
Alcohol, drink/day	9.94 ± 0.11	9.99 ± 0.01	9.96 ± 0.08	>0.05	−1.38	−0.90
Total score	39.94 ± 6.33	57.50 ± 10.78	46.97 ± 11.96	<0.05	−2.94	−1.91

**Table 4 sports-07-00044-t004:** Mean nutritional knowledge scores and correlation with Mediterranean dietary adequacy score among study participants (*n* = 10).

Nutritional Knowledge	Mean Score						Expressed as a Percentage						
	Males	Females	Total	*p*-value	*t*-statistic	Cohen’s *d*	Males	Females	Total	*p*-value	*t*-statistic	Cohen’s *d*	Correlation with MDAS
Total score	24.33 ± 5.32	33.25 ± 0.50	27.90 ± 6.08	<0.05	−4.08	−2.65	67.59 ± 14.77	92.36 ± 1.39	77.50 ± 16.89	<0.05	−4.08	−2.65	0.675*
Dietary recommendation	9.67 ± 2.34	10.50 ± 1.00	10.00 ± 1.89	>0.05	−0.773	−0.50	56.86 ± 13.75	61.76 ± 5.88	58.82 ± 11.09	>0.05	−0.77	−0.50	0.420
Nutrient sources	7.33 ± 4.41	11.00 ± 0.82	8.80 ± 3.82	>0.05	−1.99	−1.29	56.41 ± 33.94	84.62 ± 6.28	67.69 ± 29.41	>0.05	−1.99	−1.29	0.399
Healthy meals	2.50 ± 1.52	3.25 ± 0.50	2.80 ± 1.22	>0.05	−1.12	−0.73	62.50 ± 37.91	81.25 ± 12.50	70.00 ± 30.73	>0.05	−1.12	−0.73	0.170
Diet−disease association	4.83 ± 1.33	8.50 ± 0.58	9.00 ± 6.30	<0.05	−5.97	−3.88	34.52 ± 9.49	60.71 ± 4.12	45.00 ± 15.45	<0.05	−5.97	−3.88	0.730*

* Correlations are significant at *p* < 0.05.

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
