# Peer review of "Nutritional Knowledge and Dietary Practice in Elite 24-Hour Ultramarathon Runners: A Brief Report"

_sports, 2019, doi:10.3390/sports7020044_

Round 1
Reviewer 1 Report
This study was designed to assess the nutritional education and adequacy of ultramarathon runners. The concept is interesteing, but the paper does seem to lack in some significant areas. The rationale for the study is sound as there are limited data on regular dietary practices for ultra-endurance athletes, and nutrition is a critical component of trainng and racing.
Major concern(s):
However, the authors only used one limited tool to assess diet quality, when there are several options. It might be viable to use the Mediterranean diet survey, but it is not clear if the portion assumptions are valid with ultra-endurance athletes — given their energy expenditure training and racing, a difference in portion estimates will greatly impact the dietary adequacy score. Likewise, there is no assessment, or estimation, of daily intake (energy intake, macronutrient/micronutrient intakes, etc.), which would be very beneficial for the reader to know what else, and in what amounts, the runners eat regularly. With such small size, this type of assessment should be considered, as a dietary recall or food diary is not an excessive burden for the participant or technician/scientist.
Additionally, the dietary adequacy score is based on the Mediterranean diet, while other dietary approaches might view ‘adequacy’ differently. It is recommended to refer to the data as ‘Mediterranean Dietary Adequacy Score’.
Minor issues:
There are a few grammatical errors (comma placement, use of past-tense, run-on sentences) that need to be addressed. However, the paper was generally well-written from a grammar/syntax perspective.
Overall, the topic is of interest but the study design is lacking rigor. A more thorough assessment would significantly improve the the merit of this study.
Author Response
Reviewer 1
This study was designed to assess the nutritional education and adequacy of ultra marathon runners. The concept is interesting, but the paper does seem to lack in some significant areas. The rationale for the study is sound as there are limited data on regular dietary practices for ultra-endurance athletes, and nutrition is a critical component of training and racing.
Major concern(s):
However, the authors only used one limited tool to assess diet quality, when there are several options. It might be viable to use the Mediterranean diet survey, but it is not clear if the portion assumptions are valid with ultra-endurance athletes — given their energy expenditure training and racing, a difference in portion estimates will greatly impact the dietary adequacy score. Likewise, there is no assessment, or estimation, of daily intake (energy intake, macronutrient/micronutrient intakes, etc.), which would be very beneficial for the reader to know what else, and in what amounts, the runners eat regularly. With such small size, this type of assessment should be considered, as a dietary recall or food diary is not an excessive burden for the participant or technician/scientist.
Response: We thank the reviewer for the valuable comment, in which we completely agree, but we are not able to change the used tool in the already registered and approved protocol under the IRB. However we included the raised comments in the limitations. For sure we will take into account your valuable comments in future studies.
Additionally, the dietary adequacy score is based on the Mediterranean diet, while other dietary approaches might view ‘adequacy’ differently. It is recommended to refer to the data as ‘Mediterranean Dietary Adequacy Score’.
Response: Changed as suggested. We now refer to “Dietary Adequacy Score” as “Mediterranean Dietary Adequacy Score” in the entire manuscript.
Minor issues:
There are a few grammatical errors (comma placement, use of past-tense, run-on sentences) that need to be addressed. However, the paper was generally well written from a grammar/syntax perspective.
Response: Done as suggested.
Overall, the topic is of interest but the study design is lacking rigor. A more thorough assessment would significantly improve the merit of this study.
Response: We thank the Reviewer for the valuable comments.
Reviewer 2 Report
Major comments
Effect size calculations and t statistics for all tests should be presented either in the text or in tables. This will help clarify what analysis resulted in the observed p values. Since both t tests and correlations are presented for some data these statistical results are not clear.
Dietary practices in athletes may differ based on training cycle. Were these athletes in peak training, off-season, recovery from a major race, etc. at the time of this study? This can provide some context for the results.
The introduction stresses the importance of education programs to increase knowledge and hopefully see an increase in dietary adequacy. However, this is not directly addressed in the discussion and conclusion. The final paragraph should state the authors’ conclusion on this statement based on the significant correlation between DAS and total nutrition knowledge. That is, high knowledge seems to be related to high DAS, thus education to increase knowledge may have beneficial outcomes for actual dietary practices. Though, as the authors do mention, this would need to be explored in a future study.
Specific comments
Page 1. Line 38. Athletes used to consume inadequate diets? Does this mean that all athletes now have adequate diets? Wouldn’t this statement indicate that the study is not necessary. I think the authors mean that some athletes have inadequate diets and this justifies the need for research in a specific, underrepresented sport (i.e., ultrarunners).
Page 2. Line 57. Specific, testable hypotheses should be included at the end of the Introduction.
Page 2. Lines 67-68. Include manufacturer information for the scale and stadiometer.
Page 2. Line 68. Write out Body Mass Index (BMI) before using the abbreviation.
Page 2. Line 69. Weight is a unit of force and typically measured in Newtons or pounds. Kilograms is a unit of mass.
Page 2. Line 72. What questions were included in this medical history? Table 2 includes ‘self-reported health’ but this is very vague. Were there follow-up questions or additional details as to why someone categorized themselves as ‘good’, ‘satisfactory’, or ‘poor’?
Table 1. It would be easier to read the table if all components were left aligned. Then, under the ‘Alcohol, drinks per day’ category the men/women rows can be tabbed in. This will make it more clear that these are subheadings under the Alcohol label.
Page 3. Line 92. Indicate ‘where appropriate’ at the end of this sentence to indicate that the first list (statistics) is not an item by item list corresponding to how they were used in the second list.
Page 3. Lines 91-92. Indicate which variables were used in the t-tests and chi-square tests. Indicate what the independent variable was in each case. Were these t-tests one-tailed or two tailed?
Page 3. Lines 98-100. As above in the statistics recommendation, indicate in the Results or in Table 2 that the chi-square test is comparing frequencies between men/women for the observed categories. Was the expectation for the chi-square tests that men/women would have equal frequencies of these categories? It is not clear in the table and not addressed in the text. A statement should clarify that there is no significant difference in education status for the men and women, for example.
Table 2. What were the ranges of income reported for each category?
Table 2. Similar to the above comment for Table 1, left justify all text and then tab in for subheadings within the Level of Education, Occupation, Income, and Self-Reported Health to increase readability.
Table 3. Similar to above, left justify categories. Tab in the ‘no’ and ‘yes’ for Whole Grains. Change the column widths so the category names each fit on one line.
Page 4. Line 109. Total knowledge score was evaluated with >50% satisfactory. Were any participants found to be below this threshold? If so, indicate how many. If not, make this clear. Right now, there is no reference to this threshold beyond its appearance in the methods. Include this data in the text.
Table 4. Similar to above, left justify categories. This table needs to be reformatted to avoid cutting into separate lines of text. Perhaps include the percentage scores in a row beneath the mean scores to extend the table vertically and remove the right columns, or change the orientation so text runs vertically rather than left to right.
Table 4. A note should accompany the table to clarify that the * symbol denotes a significant correlation between the total (including men and women) for the two metrics used in the correlation analysis.
Page 4. Lines 113-116. A scatterplot would be helpful here to visualize the data. Include the men/women as separate symbols given that the two groups differed in their total knowledge scores.
Page 4. Line 115. Was the p value equal to .045 or less than .045? The < is confusing given the alpha level was .05.
References. Refer to reference style expectations for the journal. Both in-text and reference lists should reflect the journal style.
Author Response
Reviewer 2
Major comments
Effect size calculations and t statistics for all tests should be presented either in the text or in tables. This will help clarify what analysis resulted in the observed p values. Since both t tests and correlations are presented for some data these statistical results are not clear.
Response: Done as suggested. Now t-statistic and effect size (Cohen’s d for t-test and Cramer’s V for X2) are now presented in tables 2,3 and 4.
Dietary practices in athletes may differ based on training cycle. Were these athletes in peak training, off-season, recovery from a major race, etc. at the time of this study? This can provide some context for the results.
Response: We specified that the reported dietary practices of our athletes were the usual dietary intake followed during the year, not those during peak training or recovery from a major race. Moreover we specified that our athletes were during their off-season period (Lines 54-55 and 68).
The introduction stresses the importance of education programs to increase knowledge and hopefully see an increase in dietary adequacy. However, this is not directly addressed in the discussion and conclusion. The final paragraph should state the authors’ conclusion on this statement based on the significant correlation between DAS and total nutrition knowledge. That is, high knowledge seems to be related to high DAS, thus education to increase knowledge may have beneficial outcomes for actual dietary practices. Though, as the authors do mention, this would need to be explored in a future study.
Response: Done as suggested, now we added the statement in the Conclusion section (Lines 178-179).
Specific comments
Page 1. Line 38. Athletes used to consume inadequate diets? Does this mean that all athletes now have adequate diets? Wouldn’t this statement indicate that the study is not necessary. I think the authors mean that some athletes have inadequate diets and this justifies the need for research in a specific, underrepresented sport (i.e., ultra runners).
Response: Done as suggested, and added “some” (Line 39).
Page 2. Line 57. Specific, testable hypotheses should be included at the end of the Introduction.
Response: Done as suggested. A specific and testable hypotheses is now included at the end of the Introduction Section (Lines 59-60).
Page 2. Lines 67-68. Include manufacturer information for the scale and stadiometer.
Response: Done as suggested (Lines 72-73).
Page 2. Line 68. Write out Body Mass Index (BMI) before using the abbreviation.
Response: Done as suggested (Line 73).
Page 2. Line 69. Weight is a unit of force and typically measured in Newtons or pounds. Kilogram is a unit of mass.
Response: We completely agree with the reviewer, however reporting body weight in Kg is widely used in medical studies.
Page 2. Line 72. What questions were included in this medical history? Table 2 includes ‘self-reported health’ but this is very vague. Were there follow-up questions or additional details as to why someone categorized themselves as ‘good’, ‘satisfactory’, or ‘poor’?
Response: The questions regard the medical history were: (i) do you suffer from hypertension, cardiovascular diseases, hypercholesterolemia, diabetes, osteoporosis, articular diseases, depression, anxiety or others diseases to specify?; (ii) If yes, do you take any medication for any of these diseases? And (iii) in general how do you consider your health status? In case of any affirmative answer on the first two questions, or a “poor” self-reported health status, a detailed visit by the IUTA medical staff was programmed (Lines 78-83).
Table 1. It would be easier to read the table if all components were left aligned. Then, under the ‘Alcohol, drinks per day’ category the men/women rows can be tabbed in. This will make it more clear that these are subheadings under the Alcohol label.
Response: Done as suggested.
Page 3. Line 92. Indicate ‘where appropriate’ at the end of this sentence to indicate that the first list (statistics) is not an item-by-item list corresponding to how they were used in the second list.
Response: Done as suggested. We added the statement “where appropriate” (Line 101).
Page 3. Lines 91-92. Indicate which variables were used in the t-tests and chi-square tests. Indicate what the independent variable was in each case. Were these t-tests one-tailed or two tailed?
Response: Done as suggested (Lines 102-105).
Page 3. Lines 98-100. As above in the statistics recommendation, indicate in the Results or in Table 2 that the chi-square test is comparing frequencies between men/women for the observed categories. Was the expectation for the chi-square tests that men/women would have equal frequencies of these categories? It is not clear in the table and not addressed in the text. A statement should clarify that there is no significant difference in education status for the men and women, for example.
Response: Done as suggested (Lines 114-115).
Table 2. What were the ranges of income reported for each category?
Response: Now we included in Table 2 the ranges of family income for each category.
Table 2. Similar to the above comment for Table 1, left justify all text and then tab in for subheadings within the Level of Education, Occupation, Income, and Self-Reported Health to increase readability.
Response: Done as suggested.
Table 3. Similar to above, left justify categories. Tab in the ‘no’ and ‘yes’ for Whole Grains. Change the column widths so the category names each fit on one line.
Response: Done as suggested.
Page 4. Line 109. Total knowledge score was evaluated with >50% satisfactory. Were any participants found to be below this threshold? If so, indicate how many. If not, make this clear. Right now, there is no reference to this threshold beyond its appearance in the methods. Include this data in the text.
Response: Done as requested (Lines 123-124).
Regarding the threshold of scoring nutrition Knowledge, a score below 50% to indicate poor knowledge is widely accepted as reported in the included reference: Webb MC, J Nutr Metab. 2014;2014:506434.
Table 4. Similar to above, left justify categories. This table needs to be reformatted to avoid cutting into separate lines of text. Perhaps include the percentage scores in a row beneath the mean scores to extend the table vertically and remove the right columns, or change the orientation so text runs vertically rather than left to right.
Response: Done as suggested, orientation changed to landscape.
Table 4. A note should accompany the table to clarify that the * symbol denotes a significant correlation between the total (including men and women) for the two metrics used in the correlation analysis.
Response: Done as suggested.
Page 4. Lines 113-116. A scatterplot would be helpful here to visualize the data. Include the men/women as separate symbols given that the two groups differed in their total knowledge scores.
Response: Done as suggested. We add figure 1 as a simple scatter plot with fit line for Mediterranean dietary adequacy score by Total Nutrition Knowledge score by sex.
Page 4. Line 115. Was the p value equal to .045 or less than .045? The < is confusing given the alpha level was .05.
Response: the p value is equal to .045 (Line 133)
References. Refer to reference style expectations for the journal. Both in-text and reference lists should reflect the journal style.
Response: Done as suggested.
Reviewer 3 Report
Nutritional knowledge and dietary practice in 2 professional 24-hour ultramarathon runners: a brief 3 report
General comments
You use the term ‘professional’ for your runners. I would judge a professional athlete when he can earn his living by money from sponsors and prize money from races. Is this the case for your runners then you can use this term. When your runners are elite runners competing at top level, so I would name then ‘elite’ and not ‘professional’.
Specific comments
Line 18: change is to was
Line 20: change seems to seemed
Line 24: change seems to seemed
Line 27: change is to was
Line 122: move table 3 to page 4
Line 175: add the implications for future research and the practical applications from your study
Author Response
General comments
You use the term ‘professional’ for your runners. I would judge a professional athlete when he can earn his living by money from sponsors and prize money from races. Is this the case for your runners then you can use this term. When your runners are elite runners competing at top level, so I would name then ‘elite’ and not ‘professional’.
Response: Done as suggested.
Specific comments
Line 18: change is to was
Response: Done as suggested.
Line 20: change seems to seemed
Response: Done as suggested.
Line 24: change seems to seemed
Response: Done as suggested.
Line 27: change is to was
Response: Done as suggested.
Line 122: move table 3 to page 4
Response: We thank the reviewer, however we think it is better to leave it on page 5 to facilitate it reading and avoid confusion with table 2.
Line 175: add the implications for future research and the practical applications from your study
Response: Our findings evidence the association between higher nutritional knowledge and better dietary practice in elite 24-hour ultramarathon runners. We suggested future studies to assess the usefulness of educational programmes as a strategy to improve the adequacy of dietary intake in this specific population (Lines 174-178).
Round 2
Reviewer 2 Report
Line 131. Single instance of DAS not replaced by MDAS
Author Response
Line 131: Single instance of DAS not replaced by MDAS
Response: Done as suggested.